# The Socioeconomic Welfare of Urban Green Areas and Parks; A Literature Review of Available Evidence

Antonios Kolimenakis, Alexandra D. Solomou *, Nikolaos Proutsos, Evangelia V. Avramidou, Evangelia Korakaki, Georgios Karetsos, Georgios Maroulis, Eleftherios Papagiannis and Konstantinia Tsagkari

Hellenic Agricultural Organization "ELGO-DIMITRA", Institute of Mediterranean and Forest Ecosystems, Terma Alkmanos, Ilisia, 11528 Athens, Greece; akolimenakis@gmail.com (A.K.); proutsos@yahoo.com (N.P.); aevaggelia@yahoo.com or avramidou@fria.gr (E.V.A.); e.korakaki@gmail.com (E.K.); karetsos@fria.gr (G.K.); georgios_maroulis@eesd.gr (G.M.); lefterispapagiannis@gmail.com (E.P.); director@fria.gr (K.T.)
* Correspondence: alexansolomou@gmail.com

**Abstract:** Urban green areas present a lucid example for the harmonious co-existence of the artificial and natural environments best illustrated by their interdependence and interconnection in urban spaces. Urban green areas are essential for the health and wellbeing of citizens. The present study aimed to investigate those multiple benefits for citizens that arise through the existence of urban green areas, as well as important policy dimensions that should be considered when designing the expansion of urban green spaces in urban development. The study was based on a literature review to examine for available evidence on the benefit levels derived by the existence of urban green areas. An extended literature review was followed by a structured review, based on specific inclusion and exclusion criteria, which partly followed the Preferred Reporting Items for Systematic Reviews and Meta-Analyses (PRISMA) guidelines. The search was conducted in two databases, and a total of 1674 articles and abstracts were identified through the database searches. After removing 114 duplicates, 1560 records were initially screened based on title and abstract. Following inclusion and exclusion criteria, 14 articles were incorporated in the structured review and a total of 47 in the extended review. The extended literature review identified 33 additional articles examining aspects of benefits that did not fall under the pre-established inclusion and exclusion criteria used in the structured review, such as health benefits and other social parameters associated with urban green spaces. The selected studies were allocated in five principal groups according to study types: three of them consisted of studies employing *"willingness to pay" (WTP)* methods, five were based on *property values*, two studies assigned *monetary values*, while another two assigned $CO_2$ *values*, and, finally, two studies were based on *qualitative criteria*. The results indicated benefits to citizens and increased welfare levels gained by the existence of urban green areas. The conducted review revealed a number of findings and recommendations that could direct future research and urban policy. Those hints could assist local authorities as well as stakeholders in order to measure and assess the benefits of green spaces and urban parks and promote measures and programs to assist their further deployment.

**Keywords:** urban policies; willingness to pay (WTP); urban green areas and parks; property values; monetary values; $CO_2$ values

## 1. Introduction

The rapid growth of urbanisation has not diminished the important role of the natural environment and ecosystems as their associated values covering a wide range of services including mental and physical regeneration, cultural/educational feedback, and even food provision. Natural ecosystems are associated with the provision of clean air, soil, and water, and they contribute to the protection against natural disasters [1–3]. This re-evaluation of natural values, and the effort to bring nature closer, has made urban green areas very important and necessary for the design of sustainable approaches [4–6].

It is noteworthy that urban green areas present a lucid example for the harmonious co-existence of the artificial and natural environments most well illustrated by their interdependence and interconnection in urban spaces [7]. This connection is of the utmost importance for maintaining and improving the well-being of citizens in the cities, while at the same time maintaining the equilibrium of their associated ecosystem services [7].

The association of urban green areas (UGAs) with public health is highly acknowledged. In a recent report of the World Health Organization [8], the effectiveness of UGAs and their impacts on human health were assessed, showing that green spaces offer significant socio-economic benefits. The recent COVID 2019 pandemic further underlined the significance and the role of UGAs and urged the need for re-thinking the design and organization of modern cities to more resilient and sustainable schemes. The re-emergence of the concept of the "15-min city" proposed in 2016 [9] and its implementation in many cities around the word was accelerated because of the pandemic, proposing urban planning based on the proximity to urban green areas and relevant facilities [10].

Several case studies calculate the impact of urban green areas on citizens' welfare by using the "hypothetical market" method. The reason is quite obvious since in many cases the absence of a structured market is substituted by the creation of a dummy market, which expresses those indirect positive effects such as carbon sequestration and improved quality of life through the willingness-to-pay or other methods [11]. Additionally, the increased value of properties located near the area of green urban spaces can be regarded as another indirect monetary indicator that hints towards the benefits derived from urban green areas.

Such estimations can be of primary importance for the design and governance of urban green areas and for further monitoring and integrating other significant parameters, which are: increased urbanization, increased population density, and economic constraints on the public and municipal budget that could impact the overall citizens' benefit derived by the development and expansion of urban green areas [3,12].

The present study aimed to investigate those multiple benefits for citizens that arise through the existence of urban green areas, as well as important policy options that should be considered when deciding the development of urban green areas. The study was based on an extended literature review and a structured review based on inclusion and exclusion criteria to examine available evidence on the benefit levels derived by the existence of urban green areas.

## 2. Materials and Methods

This structured review partly followed the PRISMA guidelines for conducting systematic reviews and meta-analyses [13]. The literature search was carried out between October and December 2020 in two databases: Google Scholar and Scopus, which are two major databases with freely accessible articles related to socioeconomic analysis. The search was initially conducted on full titles and abstracts, without any geographical, date, or language limits in the selected databases. The cited references of all articles were also reviewed for potentially eligible studies. Data extraction was completed in a predefined data extraction sheet. The eligibility criteria for the assessed literature were: "*Urban green space*" OR "*Urban green areas*" OR "*Urban parks*" OR "*Urban ecosystems*" AND "*economic value*" OR "*economic benefits*" OR "*ecosystem services*" OR "*monetary values*" OR "*carbon monetary values*" OR "*climate change*". Searches were conducted with a combination of different keywords in the two databases.

The pre-established inclusion criteria included: (i) studies focusing on socioeconomic assessment relevant to the accrued benefits of urban green spaces and parks and (ii) studies assigning monetary values relevant to the social welfare of urban green spaces and parks. The main exclusion criteria included: (i) those studies with no access in the full text or study language if different from English, (ii) studies containing only a theoretical/descriptive approach of the socio-economic benefits and not providing numerical values, and (iii) studies not focusing on urban green spaces and parks.

For the extended review, the same search strategy was followed to identify articles examining additional aspects of benefits related to the existence of urban green areas and parks, which did not fall under the pre-established inclusion and exclusion criteria used in the structured review.

## 3. Results

### 3.1. Study Selection

A total of 1674 articles and abstracts were identified through the two database searches. After removing 114 duplicates, 1560 records were initially screened based on title and abstract. Two reviewers (AK and LP), working independently, reviewed the titles, abstracts, and contents of these articles. They initially excluded 1128 articles that did not meet the initial inclusion criteria and a further 231 articles according to the rest of the inclusion and exclusion criteria. A total of 14 articles were incorporated in the structured review (Figure 1: PRISMA flowchart). The review was enriched with the addition of a further 33 articles that were chosen because of the relevance of their content, but which did not fit all the inclusion and exclusion criteria. These additional articles (*n* = 33) as well as the articles of the structured review (*n* = 14) were the main body of studies of the extended review (*n* = 47). These 33 articles were also identified through the same databases.

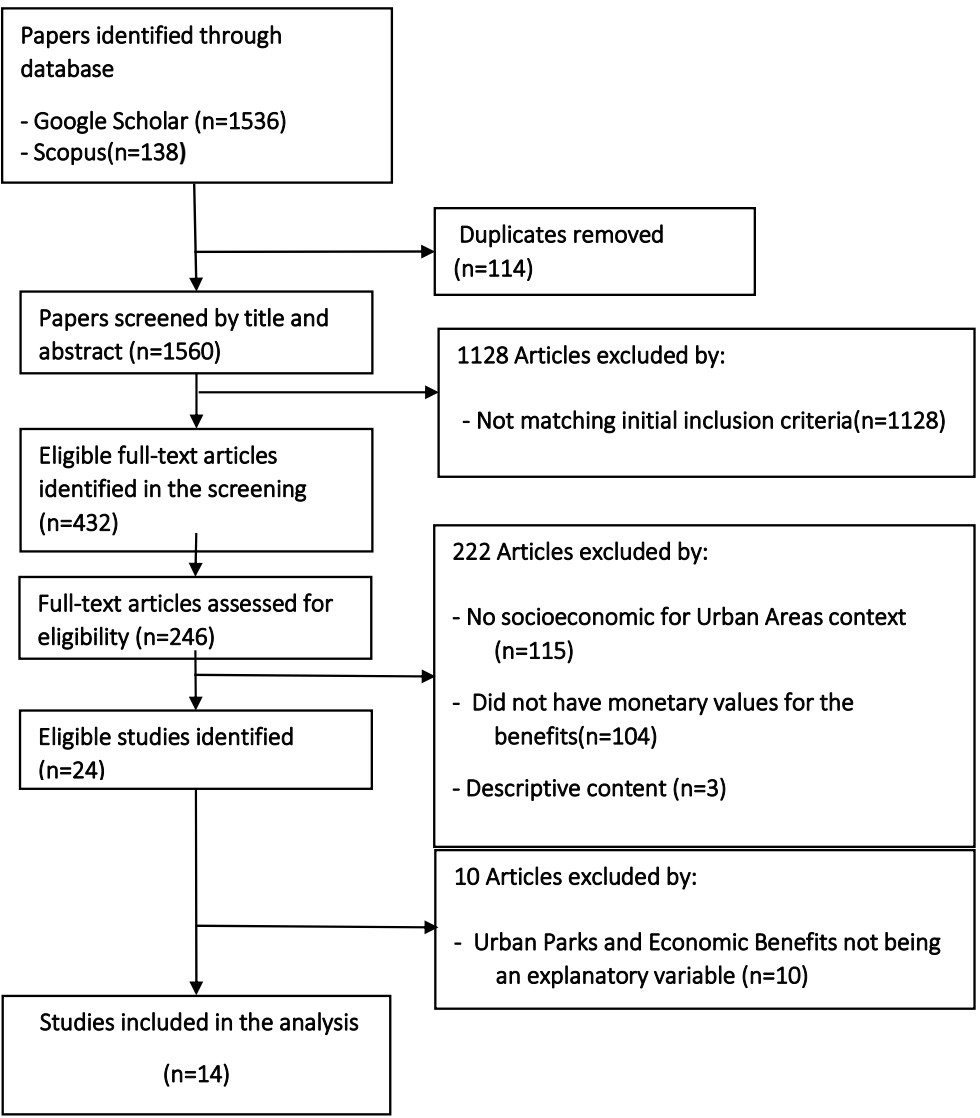

**Figure 1.** PRISMA flowchart.

### 3.2. General Study Characteristics of the Structured Review

All 14 [14–27] articles were published between 1999 and 2020. Among the included articles, four were conducted in Asia, four in North America, four in Europe, and one in Africa, while one of them was conducted in wider geographical contexts. Selected articles were further allocated in five principal groups according to study types: three consisted of studies employing *"willingness to pay (WTP)"* methods, five were based on *property values*, two were assigned *monetary values*, while two were assigned *$CO_2$ values*, and two were based on *qualitative criteria* (Table S1, Supplementary Materials). Studies were then divided according to their policy influence factor in two categories: *planning and investment for urban green areas and parks* (*n* = 7) and *the role of urban green areas and parks for improving ecosystem services* (*n* = 7) (Table 1). According to each study's type and design, the included articles were divided into two major categories: a) observational-analytic, for those studies employing a survey design, or analysis of property values, or analysis of $CO_2$ values, or analysis of health or social benefits related to urban green areas and b) reviews, for those studies based on a review of literature evidence.

### 3.3. Analysis of Evidence

3.3.1. Extended Literature Review

Prior to the description of the structured review, a brief overview of the extended literature review is presented (Table 2). The extended literature review identified 33 additional articles (28–60). All articles presented traits and benefits of urban green areas and parks. However, they did not fit to the criteria defined for the structured review. More specifically, they lacked the economic assessment, and none of the analyses provided monetary values. Nevertheless, their results and conclusions are of interest and can provide insights for further research.

The first group of studies [28–38] focused on the health benefits that urban green spaces provide. One study revealed the that urban green spaces in Sheffield (UK) have a beneficial health effect, despite the fact that a causal relationship was difficult to establish [28]. Proximity to parks have also be proved to be beneficial as it urged residents near parks to exercise more (40% more) [29]. In contrast, high levels of stroke mortality were observed in areas with lower levels of exposure to green space [30]. This effect can be correlated by two further studies carried out in the Netherlands [31,32] as the amount of green space present in the respondents' living environments had a positive relationship with their perceived general health [31], while the annual prevalence rates of 15 of 24 examined disease clusters were lower in areas with more green within a 1 km radius [32]. The research findings of the second study in the Netherlands [32] revealed that the urban green spaces have a more direct positive effect on the health of young children, while the benefits to children with attention deficit disorder were also apparent in a study in the US [33]. Further beneficial effects, especially on residents' well-being were also observed in several studies. More specifically, a study carried out in Denmark [34] showed that more frequent use of urban green space was correlated with less reported stress. Apart from that, proximity to green space was also associated with better self-reported health. Two studies conducted in the Netherlands [35,36] revealed the great contribution of urban green spaces in citizen's emotional well-being as they are less affected by stress. A further study in the US quantified this effect as an equivalent of the decrease in the local unemployment rate by 2% [37]. Additionally, a study conducted in Japan [38] provided evidence for the physiological (lower heart rate, higher parasympathetic nerve activity) and psychological relaxation effects (significantly lower levels of negative emotions and anxiety) of walking in urban parks during autumn. Apart from that, a study conducted in three urban parks in Rome [39] confirmed that the sound environment in urban parks is often positively considered, despite the fact that noise levels are above average. This is due to the influence of other factors (presence of trees, natural features, and the tranquillity) and directly affect a person's psychological state.

**Table 1.** Summary table of structured review.

| Author Year | Year | Study Location | Study Type | Study Design | Methods | Urban Green Space Context | Measured Outcome | Key Findings & Conclusions | Influence Factor |
|---|---|---|---|---|---|---|---|---|---|
| Mella et al. (2016) | 2016 | Sheffield, UK | Observational-analytic (OA) | Willingness to pay (WTP) | (WTP) survey ($n = 510$) for investments using 3D visualisations of three alternative urban greening scenarios. | Generally UGS | The influence of green infrastructure on aesthetic quality, functionality, and amenity. | The study concluded that investment in urban GI that is visibly greener, that facilitates access to GI and other amenities, and that is perceived to promote multiple functions and benefits on a single site (i.e., multi-functionality) generates higher WTP values. | Planning and Investment for Urban Green Spaces and Parks |
| D' Acci (2014) | 2014 | Turin, Italy | Observational-analytic (OA) | Property values | The result of a willingness-to-pay survey, and the definition of positional value. | Generally UGS | Urban quality of life is a hierarchical multi-attribute concept whose attributes can be defined and evaluated by several kinds of methods such as monetary (hedonic price, willingness-to-pay, cost-benefit, positional value), subjective (life satisfaction, subjective wellbeing, ranking/rating evaluation) and quantitative (how many urban attractions there are in the city and how they are distributed on its planimetry). | The magnitude increase in urban quality of life studies was directly connected with the increase in the urban population in the world. Everybody knows what makes city life better (parks, gardens, pedestrian areas, buildings and street beauty, cultural stimulation, good public transportation systems … ) and worse (crime, congestion, pollution, anonymity, alienation … ). More and more often it is appearing necessary to measure this amount of pleasentess/unpleasentess, and to assess its determinants, in order to provide a concrete value to use inside policies and investment decisions. | Planning and Investment for Urban Green Spaces and Parks |
| Molla (2015) | 2015 | North West America, California, USA | Review (RE) | Property values | Review based on studies for economic benefits of green infrastructure. | Generally UGS | Opportunities for creating new jobs, limiting the environmental impact of towns and cities, and reducing the costs of running them. | Urban public parks can help in increasing the property value; the real estate market consistently demonstrates that many people are willing to pay a higher amount for a property located close to parks or open space areas than for homes that do not offer this facility. A park basically becomes one of a city's landmarks and attractiond, making it a prime marketing tool to attract tourists, conventions, and business. | Planning and Investment for Urban Green Spaces and Parks |

**Table 1.** *Cont.*

| Author Year | Year | Study Location | Study Type | Study Design | Methods | Urban Green Space Context | Measured Outcome | Key Findings & Conclusions | Influence Factor |
|---|---|---|---|---|---|---|---|---|---|
| Fausold et al. (1999) | 1999 | USA | Review (RE) | Property values | This article reviewed different concepts of economic value in relation to open space, described methods for quantifying these values, and presented examples of each from published literature. | Generally UGS | Economic impacts that open space lands have on local communities and economies include fiscal impacts on municipal budgets, expenditures from open space-related activities, and impacts from employment and tax revenues. | A comprehensive consideration of the multiple values of open space will better inform community decisions about land conservation and development. | Planning and Investment for Urban Green Spaces and Parks |
| Millward et al. (2011) | 2011 | Toronto, Canada | Observational-analytic (OA) | Benefits of treed areas | Street Tree Resource Assessment Tool for Urban Forest Managers (STRATUM) to investigate the value of services provided by trees in Allan Gardens, a historic public park in downtown Toronto, Canada. | A public park | Quantifying the value of nature in public city spaces, in the form of treed parks. | Treed urban parks provide numerous social, environmental, and economic services of measurable value to a city. | Planning and Investment for Urban Green Spaces and Parks |
| Yeshitela (2020) | 2020 | Sub-Saharan, Addis Ababa, Ethiopia, Africa | Observational-analytic (OA) | Household survey | Data were collected through a household survey (*n* = 398) and three focus group discussions. Data were analyzed using descriptive statistics and chi-square tests. | Generally UGS | Understand the perceptions and attitudes of residents towards the benefits, challenges, and qualities of neighborhood parks in a formal settlement area in Addis Ababa, Ethiopia. | This study highlighted the importance of place-based studies for assessing the perceived benefits that attract people to use urban parks, as well as the challenges that deter use. | Planning and Investment for Urban Green Spaces and Parks |
| Sim (2020) | 2020 | Gyeongui Line Forest Park in Seoul, South Korea | Observational-analytic (OA) | Economic benefits on local business from an urban park | The author divided the linear park into five sections according to each section's design strategy to examine the relationship between the design features and card transaction behaviors. | Urban Parks | This study fills the gap between big data and park studies by using card transaction data within 400 m of the Gyeongui Line Forest Park in terms of economic benefits on local business. | Compared to other sections, only Section 1, as an open space within a commercialized area, contributed to local business positively. Other sections, such as 2–5, represented the negative impacts on local business from 2016 to 2017. | Planning and Investment for Urban Green Spaces and Parks |
| Lo et al. (2010) | 2010 | Hong Kong | Observational-analytic (OA) | WTP | A total of 495 urban residents from different neighbourhoods and socio-economic groups were interviewed about how often they visit an urban park. | An urban park in Hong Kong | This study investigated Hong Kong residents' recreational use of urban green spaces and assessed the monetary value of these areas. | People attach multiple values to urban green spaces, which play varied roles in cities. | The role of UGS and Parks for improving Ecosystem Services |

**Table 1.** *Cont.*

| Author Year | Year | Study Location | Study Type | Study Design | Methods | Urban Green Space Context | Measured Outcome | Key Findings & Conclusions | Influence Factor |
|---|---|---|---|---|---|---|---|---|---|
| Sirina et al. (2017) | 2017 | France | Observational-analytic (OA) | WTP | The probability of park users' willingness to pay (WTP) to enjoy an urban park in a medium-sized conurbation in the French context. A questionnaire to obtain a quick glance at how local governments and city planners see the benefits of parks. | An urban park in a medium-sized conurbation | Willingness to pay (WTP) to enjoy an urban park. | We see how local governments and city planners see the benefits of parks. | The role of UGS and Parks for improving Ecosystem Services |
| Gratani et al. (2016) | 2016 | Rome, Italy | Observational-analytic (OA) | $CO_2$ values | $CO_2$ sequestration capability of the vegetation developing in parks of four historical residences (Villa Pamphjli, Villa Ada Savoia, Villa Borghese, and Villa Torlonia) in Rome and its economic value were analyzed. | Four UGS in Rome | Economic value for $CO_2$ sequestration. | Urban parks form the largest proportion of public green spaces contributing to both physical and mental well-being of people living in urban areas. | The role of UGS and Parks for improving Ecosystem Services |
| Gao et al. (2011) | 2011 | Shenyang, China | Observational-analytic (OA) | $CO_2$ values | To better understand the importance of the urban parks, by using Quickbird satellite image (QB) and also adopting a Geographic Information System (GIS), we employed the methods of image interpretation, site survey, photosynthetic experiment, and forestation costs method, assessing the quantities and monetary benefits of vegetations in urban parks. | Generally UGS | As cities around the word increasingly focus on achieving greater energy efficiency, reducing pollution, and promoting greater long-term sustainability, the kind of analysis we showcase here can inform and support these efforts. | Urban parks are one amenity that provide cities with value in the form of environmental services and enhanced aesthetic beauty. | The role of UGS and Parks for improving Ecosystem Services |
| Sadeghian et al. (2013) | 2013 | Chicago, USA, Singapore and Kuala Lumpur, Malaysia | Review (RE) | Property values | Major literature review relating to the benefits of urban parks. | Generally UGS | Economic benefits: energy savings, urban parks and water management, and property value. | Urban parks have many functions and benefits. These functions and benefits are important to improve life quality in the urban areas. | The role of UGS and Parks for improving Ecosystem Services |

**Table 1.** *Cont.*

| Author Year | Year | Study Location | Study Type | Study Design | Methods | Urban Green Space Context | Measured Outcome | Key Findings & Conclusions | Influence Factor |
|---|---|---|---|---|---|---|---|---|---|
| Hoover et al. (2020) | 2020 | Omaha, Nebraska, USA | Observational-analytic (OA) | Property values | Using repeat-sales models and sales data from 2000 to 2018, we evaluated changes in property values for homes within various buffers of GI-parks. Surveys and interviews of residents can be used to determine how homeowners perceive GI and their understanding of its functions and benefits. | Various buffers of GI-parks | What effects the installation of GI had on residential property values in the City of Omaha. | This finding is consistent with the notion that homeowners place little value on modifications to existing greenspace but may also stem from homeowners' lack of familiarity with GI practices or data limitations. | The role of UGS and Parks for improving Ecosystem Services |
| Yotapakdee et al. (2018) | 2018 | Bang Kachao Green Space, Thailand | Observational-analytic (OA) | Benefits of treed areas | Data was collected from six types of tree habitat classified as road-side, abandoned area, public area, private area, temple area, and park, located across six subdistricts of Bang Kachao. | Open spaces | Evaluation of the benefits of large trees in the urban area at Bang Kachao Green Space, Samutprakan province. | The recommendations based on this study will help develop appropriate policies for sustaining ecosystem services and contributions to human wellbeing. | The role of UGS and Parks for improving Ecosystem Services |

**Table 2.** Summary table of extended review.

| Author/Year | Year | Study Location | Study Type | Study Design | Urban Green Space Context | Key Findings & Conclusions |
|---|---|---|---|---|---|---|
| Mella et al. (2016) | 2016 | Sheffield, UK | Observational-analytic (OA) | WTP | Generally UGS | The article concluded that investment in urban GI that is visibly greener, that facilitates access to GI and other amenities, and that is perceived to promote multiple functions and benefits on a single site (i.e., multi-functionality) generates higher WTP values. |
| D' Acci (2014) | 2014 | Turin, Italy | Observational-analytic (OA) | Property values | Generally UGS | The magnitude increase in urban quality of life studies is directly connected with the increase in the urban population in the world. Everybody knows what makes city life better (parks, gardens, pedestrian areas, buildings and street beauty, cultural stimulation, good public transportation systems . . . ) and worse (crime, congestion, pollution, anonymity, alienation . . . ). More and more often it is appearing necessary to measure this amount of pleasentess/unpleasentess, and to assess its determinants, in order to provide a concrete value to use inside policies and investment decisions. |
| Molla (2015) | 2015 | North West America, California, USA | Review (RE) | Property values | Generally UGS | Urban public parks can help in increasing the property value with the real estate market consistently demonstrating that many people are willing to pay a higher amount for a property located close to parks or open space areas than for homes that do not offer this facility. A park basically becomes one of a city's landmarks and attractions, making it a prime marketing tool to attract tourists, conventions, and business. |
| Fausold et al. (1999) | 1999 | USA | Review (RE) | Property values | Generally UGS | A comprehensive consideration of the multiple values of open space will better inform community decisions about land conservation and development. |

**Table 2.** *Cont.*

| Author/Year | Year | Study Location | Study Type | Study Design | Urban Green Space Context | Key Findings & Conclusions |
|---|---|---|---|---|---|---|
| Millward et al. (2011) | 2011 | Toronto, Canada | Observational-analytic (OA) | Benefits of treed areas | A public park in Toronto | Treed urban parks provide numerous social, environmental, and economic services of measurable value to a city. |
| Yeshitela (2020) | 2020 | Sub-Saharan, Addis Ababa, Ethiopia, Africa | Observational-analytic (OA) | Household survey | Generally UGS | This study highlighted the importance of place-based studies for assessing the perceived benefits that attract people to use urban parks, as well as the challenges that deter use. |
| Sim (2020) | 2020 | Gyeongui Line Forest Park in Seoul, South Korea | Observational-analytic (OA) | Economic benefits on local business from an urban park | Urban parks | Compared to other sections, only Section 1, as an open space within a commercialized area, contributed to local business positively. Other sections, such as 2–5, represented the negative impacts on local business from 2016 to 2017. |
| Lo et al. (2010) | 2010 | Hong Kong | Observational-analytic (OA) | WTP | An urban park in Hong Kong | People attach multiple values to urban green spaces, which play varied roles in cities. |
| Sirina et al. (2017) | 2017 | France | Observational-analytic (OA) | WTP | An urban park in a medium-sized conurbation in the French context | We see how local governments and city planners see the benefits of parks. |
| Gratani et al. (2016) | 2016 | Rome, Italy | Observational-analytic (OA) | $CO_2$ values | Four UGS in Rome | Urban parks form the largest proportion of public green spaces contributing to both physical and mental well-being of people living in urban areas. |
| Gao et al. (2011) | 2011 | Shenyang, China | Observational-analytic (OA) | $CO_2$ values | Generally UGS | Urban parks are one amenity that provide cities with value in the form of environmental services and enhanced aesthetic beauty. |
| Sadeghian et al. (2013) | 2013 | Chicago, USA, Singapore and Kuala Lumpur, Malaysia | Review (RE) | Property values | Generally UGS | Urban parks have many functions and benefits. These functions and benefits are important to improve life quality in urban areas. |
| Hoover et al. (2020) | 2020 | Omaha, Nebraska, USA | Observational-analytic (OA) | Property values | Various buffers of GI-parks | This finding is consistent with the notion that homeowners place little value on modifications to existing greenspace but may also stem from homeowners' lack of familiarity with GI practices or data limitations. |

Table 2. *Cont.*

| Author/Year | Year | Study Location | Study Type | Study Design | Urban Green Space Context | Key Findings & Conclusions |
|---|---|---|---|---|---|---|
| Yotapakdee et al. (2018) | 2018 | Bang Kachao Green Space, Thailand | Observational-analytic (OA) | Benefits of treed areas | Open spaces | The recommendations based on this study will help develop appropriate policies for sustaining ecosystem services and contributions to human wellbeing. |
| Lee et al. (2010) | 2010 | Sheffield, UK | Review (RE) | Health benefits of urban green spaces | Generally UGS | Most studies reported findings that generally supported the view that green space has a beneficial health effect. Establishing a causal relationship is difficult, as the relationship is complex. Simplistic urban interventions may therefore fail to address the underlying determinants of urban health that are not remediable by landscape redesign. |
| Leeuwen et al. (2010) | 2014 | USA | Observational-analytic (OA) | Benefits of treed areas | Generally UGS | The post-productivist model opens perspectives to many new farm types and new urban garden forms in cities, which rapidly change their general brownish cartographic colour into an exciting mix with splashes of bluish-green. |
| Harnik et al. (2014) | 2014 | USA | Observational-analytic (OA) | Economic value of a park system to a community | Urban parks | There is an urgent need for more sophistication and better nuanced tools, but the field does not have the luxury of delaying the use of tools and evidence until they have been perfected. If scientists, economists, and advocates fail to come forward with measuring tools for the policy debate, then the parks field will be substantially disadvantaged since other competing services are not hesitant to use such measures to support their cases. |
| Loures et al. (2007) | 2007 | Portimão, Portugal | Observational-analytic (OA) | Urban parks and sustainable city planning | City Park of Portimão | The Master Plan of Portimão City Park encourages a broad range of recreational, leisure, and public uses that utilise the available facilities and infrastructure and that add to the unique qualities of the park for visitors, workers, and residents. |

**Table 2.** *Cont.*

| Author/Year | Year | Study Location | Study Type | Study Design | Urban Green Space Context | Key Findings & Conclusions |
|---|---|---|---|---|---|---|
| Allen et al. (1985) | 2009 | USA | Observational-analytic (OA) | Property values | Urban parks | We conclude that the practical problems in valuing urban parks may preclude the use of any single method of measurement. |
| Zoest et al. (2014) | 2014 | Dutch TEEB Cities | Observational-analytic (OA) | Economic benefits of green space | Generally UGS | While the project was successful in producing the desired deliverables (a tool for inclusive finance for urban green spaces, eight in-depth cases showing green spaces paying their way, and a community of practice), it is recognized that the adoption of inclusive finance in municipalities depends critically on urban strategies that have efficiency and resilience at their core. |
| TyrvaÈinen et al. (1998) | 1998 | Joensuu, the capital of North Carelia, Finland | Observational-analytic (OA) | Economic value of urban forest amenities | Urban parks | The results suggest that most visitors were willing to pay for the use of wooded recreation areas. Furthermore, approximately half of the respondents were willing to pay to prevent the conversion of forested parks to another land-use. The results can be used to assess the efficiency of urban forests management. In addition, the results are useful in assessing the value of green space benefits in different land-use options. |
| Cohen et al. (2007) | 2007 | USA | Observational-analytic (OA) | Benefits of urban green spaces | 8 urban parks | Residential proximity was strongly associated with park use and physical activity. People living within a mile of a park were four times more likely to use it once a week or more and had 38% more exercise sessions per week than those living further away. |
| Hu et al. (2008) | 2008 | USA | Observational-analytic (OA) | $CO_2$ values | Urban parks | High levels of stroke mortality were observed in areas with lower levels of exposure to green space. |
| Kweon et al. (1998) | 1998 | USA | Observational-analytic (OA) | Benefits of treed areas | Inner-city neighbourhood, USA | Exposure to green common spaces wasassociated with better social integration of elderly persons. |

**Table 2.** *Cont.*

| Author/Year | Year | Study Location | Study Type | Study Design | Urban Green Space Context | Key Findings & Conclusions |
|---|---|---|---|---|---|---|
| Lee et al. (2008) | 2008 | USA | Observational-analytic (OA) | Benefits of treed areas | 82 urban neighbourhoods | Women with low incomes or living in deprived neighbourhoods had less access to physical activity resources (including parks). Greater availability of physical activity resources nearby appeared to benefit women living in more deprived neighbourhoods and low-income women more. |
| Maas et al. (2006) | 2006 | Netherlands | Observational-analytic (OA) | Health benefits of urban green spaces | Urban, mixed urban–rural, and rural | Reported that the amount of green space present in the respondents' living environments was positively associated with their perceived general health. |
| Maas et al. (2009) | 2009 | Netherlands | Observational-analytic (OA) | Health benefits of urban green spaces | Urban areas in Holland | The annual prevalence rates of 15 of 24 disease clusters were lower in areas with more green space within a 1 km radius. The relationship was particularly strong for children and the lower socioeconomic classes. However, the effect size was small (OR: 0.95–0.98). |
| Stigsdotter et al. (2010) | 2010 | Denmark | Observational-analytic (OA) | Health benefits of urban green spaces | Urban parks | A greater use of green space was associated with less reported stress. Closer proximity to green space was also associated with better self-reported health. |
| Taylor et al. (2001) | 2001 | USA | Observational-analytic (OA) | Health benefits of urban green spaces | Urban parks | Children with attention deficit disorder function better after activities in a green setting. |
| van den Berg et al. (2010) | 2010 | Netherlands | Observational-analytic (OA) | Health benefits of urban green spaces | Urban areas in Holland | Respondents with higher levels of green space reported being less affected by stressful life events and reported better perceived mental health. |
| Bauman et al. (2007) | 2007 | Northe America & Australia | Review (RE) | Benefits of treed areas | Urban parks | Consistent associations between access, perceived safety, and aesthetic features of parks and physical activity. |

**Table 2.** *Cont.*

| Author/Year | Year | Study Location | Study Type | Study Design | Urban Green Space Context | Key Findings & Conclusions |
|---|---|---|---|---|---|---|
| Chiesura (2004) | 2004 | Netherlands | Observational-analytic (OA) | Benefits of treed areas | Amsterdam urban park | The issues investigated concern people's motives for urban nature, the emotional dimension involved in the experience of nature, and its importance for people's general well being. Results confirmed that the experience of nature in urban environment is source of positive feelings and beneficial services, which fulfill important immaterial and non-consumptive human needs. |
| More et al. (1988) | 1988 | USA | Review (RE) | Benefits of treed areas | Urban parks | Study results indicated that landscape planners need to be aware of the strengths and shortcomings of each to properly evaluate research on this topic. |
| Peters et al. (2010) | 2010 | Netherlands | Observational-analytic (OA) | Benefits of urban green spaces | Five urban parks in Netherlands | The design of a park, its location, and people's image of the park in combination with the cultural characteristics of various ethnic groups inform the opportunities for intercultural interactions. |
| Oh et al. (2007) | 2007 | South Korea | Observational-analytic (OA) | Economic value of a park system to a community | Urban parks in Seoul | Considering the actual locations of parks and the corresponding local population and land use, the approach conducted in this study provided practical ways of understanding and managing the spatial distribution of urban parks. |

**Table 2.** *Cont.*

| Author/Year | Year | Study Location | Study Type | Study Design | Urban Green Space Context | Key Findings & Conclusions |
|---|---|---|---|---|---|---|
| Granz et al. (2004) | 2004 | USA | Observational-analytic (OA) | Benefits of urban green spaces | Urban parks | Part I describes a fifth model, the sustainable park, which began to emerge in the late 1990s. Part II postulates three general attributes of this new kind of park: (1) self-sufficiency in regard to material resources and maintenance, (2) solving larger urban problems outside of park boundaries, and (3) creating new standards for aesthetics and landscape management in parks and other urban landscapes. It also explores policy implications of these attributes regarding park design and management, the practice of landscape architecture, citizen participation, and ecological education. |
| Nordh et al. (2009) | 2009 | Skandinavia | Observational-analytic (OA) | Benefits of treed areas | Small urban parks (pocket) | Formal mediation analyses indicated distinctive patterns of full and partial mediation of the relations between environmental components and restoration likelihood by being away and fascination. Our results provide guidance for the design of small yet restorative urban parks. |
| Lin et al. (2014) | 2014 | Australia | Observational-analytic (OA) | Benefits of urban green spaces | Urban parks | Park users with stronger nature orientations (i) spent more time in their yard, (ii) traveled further to green spaces, and (iii) made longer visits than park visitors with weaker nature orientations. Overall, our results suggest that measures to increase people's connection to nature could be more important than measures to increase urban green space availability if we want to encourage park visitation. |

**Table 2.** *Cont.*

| Author/Year | Year | Study Location | Study Type | Study Design | Urban Green Space Context | Key Findings & Conclusions |
|---|---|---|---|---|---|---|
| Chang et a. (2014) | 2014 | Twaivan | Observational-analytic (OA) | Effects of urban parks on the local urban thermal environment | 60 urban parks | In business and other districts used mostly during daytime, it is recommended that parks and other open spaces be designed with less than 50% paved area and at least 30% trees, shrubs, and other shadings. In residential districts that are used mostly during nighttime, parks and other open spaces are recommended to be designed with more trees. Night irrigation, a measure commonly recommended for the conservation of water, is also recommended to further enhance this nighttime cooling. |
| Azcárraga et al. (2019) | 2019 | México | Observational-analytic (OA) | Benefits of urban green spaces | Nine parks in México City | Results showed a close relationship between patterns of visitor use and urban parks' components such as distance, tree abundance, safeness, playground qualities, and cleanliness. |
| Sturm et al. (2014) | 2014 | USA | Observational-analytic (OA) | Health benefits of urban green spaces | Neighborhood parks in Los Angeles | This study provided a new data point for an arguably very old question, but for which empirical data are sparse for the US. A nearby urban park was associated with the same mental health benefits as decreasing local unemployment rates by two percentage points, suggesting at least the potential of environmental interventions to improve mental health. |
| Cohen et al. (2014) | 2014 | USA | Observational-analytic (OA) | Benefits of urban green spaces | Urban parks | The findings emphasize the importance of public green spaces in the urban tissue and justify investment in these spaces in terms of sustainable development. |

**Table 2.** *Cont.*

| Author/Year | Year | Study Location | Study Type | Study Design | Urban Green Space Context | Key Findings & Conclusions |
|---|---|---|---|---|---|---|
| Lam et al. (2005) | 2005 | Hong Kong | Observational-analytic (OA) | Benefits of urban green spaces | Hong Kong urban parks and open spaces | These findings lend support to the postulation that the capability of urban parks and open spaces in dense cities to improve the urban environment is rather limited and call for a re-examination of the role of urban parks in enhancing urban livability. The findings also have implications on how urban parks in dense cities should be designed and managed. |
| Veal (2006) | 2006 | Australia | Review (RE) | Benefits of urban green spaces | Sydney urban parks and open spaces | The review concludes that the "accepted wisdom" on the non-use and decline of urban parks is questionable and contrary to available empirical evidence and that leisure studies discourses, which ignore urban parks, as a leisure sector, provide a distorted view of the equity outcomes of public leisure services as measured by patterns of usage. |
| Shuib et al. (2015) | 2015 | Germany | Review (RE) | Benefits of urban green spaces | Urban parks | The research will increase awareness among the local community groups to preserve the values and amenities of the park and its environmental setting. The outcome of this research offers essential insights on the preferences and community values towards successful urban parks. |

**Table 2.** *Cont.*

| Author/Year | Year | Study Location | Study Type | Study Design | Urban Green Space Context | Key Findings & Conclusions |
|---|---|---|---|---|---|---|
| Song et al. (2015) | 2015 | Japan | Observational-analytic (OA) | Health benefits of urban green spaces | Urban parks | The authors observed that walking in an urban park resulted in a significantly lower heart rate, higher parasympathetic nerve activity, and lower sympathetic nerve activity than walking through the city area. In subjective evaluations, participants were more "comfortable," "natural," "relaxed," and "vigorous" after a walk in the urban park. Furthermore, they exhibited significantly lower levels of negative emotions and anxiety. These findings provide scientific evidence for the physiological and psychological relaxation effects of walking in urban parks during fall. |
| Brambilla et al. (2013) | 2013 | Italy | Observational-analytic (OA) | Benefits of urban green spaces | 3 urban parks in Rome | The results confirmed that the sound environment in urban parks is often considered as "good" or "excellent" even if the sound pressure level is nearly always higher than the limits commonly used to define quiet areas. This is due to the influence of other factors, such as the presence of trees, natural features, and the tranquility; all of these components cannot be neglected in the assessment of the soundscape because they directly affect the psychological state of the person. |

The second smaller group of studies [40–43] concerned social issues and benefits to the well-being of citizens. One study in the US expressed that exposure to green urban spaces is associated with better social integration of elderly persons [40]. One study [41] showed that greater availability of physical activity resources appears to have more beneficial effects on women living in more deprived neighbourhoods with lower income women. Apart from that, urban green spaces and parks are proved to contribute to social cohesion, as it can be the meeting place of people from different cultural and ethnic backgrounds and can facilitate intercultural interactions [42] and can contribute to local community engagement [43].

Further studies include and promote the benefits associated with urban green spaces. A study in the US justified the investment in the creation of urban parks [44], while one study advocated the construction of diverse types of parks [45] and another hailed the Master Plan of Portimão City Park [46]. Furthermore, a number of studies [47–49] underlined that it is extremely difficult to find an appropriate valuation of urban green spaces and parks [47]. However, the importance of immediately finding policy options and measuring tools [48] is underscored in order to promote local community acceptance and secure the success of specific investment programmes for urban green space [49]. With regard to urban forests, a study conducted in Finland revealed an increased willingness to pay for the use of wooded recreation areas as well as for the prevention of the conversion of forested parks to another land-use [50].

Some studies revealed general associations between access, perceived safety and aesthetic features of parks and physical activity [51]; the need for being aware of the strengths and shortcomings of the area in order to proceed with an evaluation [52]; practical ways of understanding and managing spatial distribution of urban parks [53]; a model on sustainable parks [54]; and guidance on small urban parks [55]. By focusing on a specific category of visitors of urban green spaces (those with stronger relation to nature), an Australian study implicated that measures to increase people's connection to nature could be more important than measures to increase urban green space availability [56]. Another study provided recommendations on how urban green spaces and parks should be constructed in specific districts (business and other districts in contrast with residential districts), defining a ratio between paved areas and trees [57]. A correlation between visitor frequency and traits such as distance, tree abundance, safeness, playground qualities, and cleanliness was also found [58]. Contrary to other studies, a study conducted in 60 parks in Taiwan showed the decreased capability of urban green spaces to improve the urban environment [59], while another study concluded that current dominant views on non-use of urban green spaces contradicts the existing empirical evidence und undermines the role of those spaces as a leisure sector [60].

### 3.3.2. Willingness to Pay (WTP) for Urban Green Areas and Parks

Citizens' appraisal for the existence of urban green areas is often elucidated by their willingness to assign monetary values according to different study methods and characteristics.

In a study conducted in Sheffield (UK) [14], participants were willing to pay up to £10 per month in their rents for residing in a location characterized an urban green area. A study conducted in Hong Kong [15] examined residents' use value of urban green areas and assessed it in monetary terms. According to the results of the study, respondents were willing to pay a monthly average payment of approximately USD 9.90 per household in order to restore losses of urban green spaces in their residing areas. Another study conducted in France [16] indicated that the WTP of park users to enjoy an urban park was positively influenced by visit frequency and age. The authors found that marginal age changes increase the WTP by EUR 0.025, while the accessibility to sports and events lead to a WTP increase of EUR 0.732.

### 3.3.3. Property Values Close to Urban Green Areas and Parks

A study [17] conducted in Turin (Italy) indicated that the green character of an urban area can change the value of a property up to almost 140%. Another article [18] based on studying various sites in Chicago, Singapore, and Kuala Lumpur examined the relating benefits of urban parks. Results indicated that a park can change or affect the air temperature in the surrounding area. More specifically, the study in Chicago showed that increasing tree density by 10% can lead to reductions in the required household energy use by up to 10%. Results indicate that green spaces and green landscape could increase property values from 5% to 15% depending on the type of interventions.

A review article [19], based on studies conducted in north-west America, found that green infrastructure technologies can reduce the rates of flooding, which in turn could lead to an increase in property values up to 5%. Additionally, avoided flood costs could equal to almost USD 3 million, saving USD 9000 to USD 21,000 per acre. The same study indicated that urban trees can provide up to USD 1.2 million annually in environmental and property values, leading to a benefit cost ratio of USD 3.81 for every USD 1.00 spent. In addition, according to the same study [19], tourism spending relevant to green infrastructure, resulted in a direct economic impact of USD 111 million in wages and the creation of almost 5700 jobs.

Another article [20] incorporated different studies from published literature in the USA that indicate the relation of property values with open spaces such as large urban parks. A study of urban parks in Columbia, Ohio found an increase of up to 23% in the value for properties in green area proximity. The authors concluded that each hectare of urban park area could result to an added value of up to USD 6500/ha. The authors [20] also concluded that properties in proximity to greenbelts had a value of up to 32% higher than properties in a radius of 1 km away, equal to almost USD 13.75 per meter away from a greenbelt.

A study [21] conducted in the US using repeat-sales models and data from 2000 to 2018 evaluated the increase in property values for homes within various zones of green infrastructure and parks. The results indicated a non-linear correlation with distance to the nearest green zone and age.

### 3.3.4. Economic Value of $CO_2$ Sequestration in Urban Green Areas and Parks

A study [22] conducted in Rome (Italy) examined the $CO_2$ sequestration capability and the economic value of the green infrastructure (vegetation) in four historical residences. The calculated carbon sequestration for the four parks (3197 Mg $CO_2$ ha$^{-1}$year$^{-1}$) amounted to almost 4% of the total greenhouse gas emissions of Rome for 2010, which was equal to an economic value of approximately USD 23,500/ha.

A study [23] in Shenyang (China) employed Quickbird satellite image and GIS image interpretation for the assessment of the benefits of vegetation in urban parks. The results showed that the vegetation of the studied parks in Shenyang, the capital and largest city of China's northeast Liaoning Province with more than 8.2 million inhabitants, were calculated to lead to a reduction in $CO_2$ by 197,847 t annually, resulting in a monetary benefit of approximately USD 7990.53 thousand. In addition, the vegetation's release of $O_2$ by 147,149.3 t annually resulted in an estimated benefit of USD 8039.29 thousand. The monetary benefit of carbon emissions' reduction and the release of $O_2$ was estimated at USD 16,029.82 thousand. As the results indicate, vegetation in urban parks can produce an annual monetary benefit of almost USD 13,561.99 thousand.

### 3.3.5. Monetary Values of Urban Green Areas and Parks

A study [24] in Toronto (Canada) estimated that the annual benefit, in terms of environmental and aesthetic value, provided by trees in a public park was almost USD 26,000, generating a benefit-to-cost ratio of 3.4 to 1. Another study [25] in Bang Kachao (Thailand) collected data from different types of trees from different spots of the city and found that the total monetary value of trees was almost USD 23,000 per year.

### 3.3.6. Qualitative Values of Urban Green Areas and Parks

A study [26] conducted in Sub-Saharan Africa, Addis Ababa (Ethiopia), collected data through a survey process and indicated that respondents highly appraise the environmental, socio-cultural, and economic benefits provided by parks and that the socio-cultural and environmental benefits are ranked higher than the economic benefits.

Another study [27] divided the Gyeongui Line Forest Park in Seoul (Korea) into five distinct sections. These sections were divided according to a specific strategy for designing urban parks. Thereby, a relationship was examined between the design features and card transaction behaviors. Results showed that the average ages increased from 2015 to 2017. In addition, results describe a decrease in users' ages by year, and the amounts of average card transaction also increased from 2015 to 2017 continuously.

### 3.4. Policy Recommendations

According to their specific policy influence factor, studies were divided in two main categories: *planning and investment for urban green areas and parks* (*n* = 7) [14,17,19,20,24,26,27], and *the role of urban green areas and parks for improving ecosystem services* (*n* = 7) [15,16,18,21–23,25].

### 3.4.1. Planning and Investment for Urban Green Areas and Parks

A study conducted in Sheffield (UK) [14] showed that investment in urban green space can have significant effects on the real estate market through the provision of functional and greener infrastructure. In a study conducted in Turin (Italy) [17], the authors used the concepts of "Isobenefit Lines" and the "Isobenefit Orography," for describing the spatial urban amenities' distribution and quantity. The term "Isobenefit" refers to all citizens in a city having equal access to benefits and quality, and the authors concluded that urban green areas increase the urban quality of life. In a study held in north-west America, California (USA) [19], it was concluded that the improvement of green infrastructure should be prioritized and that urban public parks can help increase the property value.

Additionally, a study [20] conducted in the USA, highlighted the benefits of urban green spaces for communities, decision makers, and urban planners. The study concluded that a thorough consideration of the multiple values of green spaces can positively affect land conservation and development. Furthermore, a study conducted in Toronto (Canada) [24] provided a methodology for quantifying the value of nature in public city spaces, using the parameter of trees in parks. The study supports the conclusion that urban parks with a high percentage of trees can provide multiple socioeconomic benefits, which are useful for urban planning.

Moreover, a study [27] in Addis Ababa (Ethiopia) supports the conclusion that further investigation is needed to comprehend the community's decisions influencing the use of public parks. The study emphasized the importance of spatial studies for assessing the benefits and gaps related to the use of urban parks. Lastly, a study [26] in Gyeongui Line Forest Park in Seoul (South Korea) provided design features, which are helpful to landscape architects and urban designers and which can positively affect the economic benefits of a park.

### 3.4.2. The Role of UGAs and Parks for Improving Ecosystem Services

The findings of a study [15] conducted in Hong Kong could inform green space planning and nature conservation and highlighted the need to consider community-based approaches when designing relevant public policies. Another study [23] indicated that urban parks are an important public good that provide cities the value of environmental services and enhanced aesthetic beauty. Moreover, according to a study [18] conducted in Chicago (USA), Singapore, and Kuala Lumpur (Malaysia), urban parks are one of the most important components of cities and bear an evolving role in improving quality of life in urban areas.

In a study [22] conducted in Rome (Italy), it was shown that parks could significantly contribute to carbon sequestration. The provision of data concerning $CO_2$ sequestration

can be utilized to create "carbon databanks" run by urban authorities. Moreover, these data can be used in GIS allowing the monitoring of $CO_2$ concentration and can inform management practices to capture the welfare of the services that urban parks can provide. The study concluded that urban parks consist of the highest proportion of public green areas and can lead to both physical and mental health of urban dwellers.

In another study [16] in France, it was shown that age and sports events are important determinants, and the study showed a high tendency for willingness to pay for older age groups. Furthermore, a study [21] in Omaha, Nebraska (USA) showed how management practices can help address infrastructure—the quality of urban service—and inform urban planning needs. The value of these benefits should be prioritized when considering the benefits and costs of management initiatives.

Lastly, a study [25] conducted in Thailand supported the conclusion that green areas contribute to the conservation of biodiversity levels in urban areas. In the case of Bang Kachao, green areas have been improved through the establishment of new gardens in the city. The recommendations raised by this study [25] could help the development of the design of policies for sustaining ecosystem services and could contribute to citizens' well-being in cities.

## 4. Discussion

The present study estimated the socioeconomic welfare and the benefits resulting from the existence of urban green spaces and parks based on the evidence provided by 14 reviewed articles. An important issue raised herein is associated with the presentation of the values relevant to the social welfare resulting from urban green spaces and parks and from the employment of different methods: (a) willingness to pay; (b) an increase in the value of properties neighbouring to green urban factors and parks; and (c) the quality of carbon monetary factors. The present study served as a source of useful policy recommendations concerning the development of urban green areas and parks based on the evidence of relevant studies.

The 14 included studies of the structured review employed a wide variety of different methodologies, indicating the multiple socioeconomic aspects related to urban green areas and the need to consider multiple factors when designing effective urban green policies. Specifically, the employment of willingness to pay methods indicated a monthly benefit level of up to USD 12. Moreover, when considering property values near a green urban area, they can increase up to 143%. In regards to the economic value of $CO_2$ sequestration in urban green areas and parks, it was estimated that this can reach an annual economic value of up to approximately USD 23,500/ha.

As presented in the extended review, there is a plethora of studies aiming to illustrate the overall benefits achieved by the existence of urban green areas [28–60]. However, it is often hard to depict those multiple effects of urban green areas on citizens' well-being in single terms and units or through the employment of monetary values. This is due to the fact that urban green areas do not provide only direct economic benefits but are also associated with a bundle of indirect benefits. Urban green infrastructure can have positive effects on citizens' well-being through the provision of essential ecosystem services [1], thus improving their quality-of-life indicators. In addition, ecosystems services such as carbon sequestration (regulating ecosystem service) or even the aesthetic pleasure (cultural ecosystem service) are factors that indirectly influence citizens' well-being and should be taken into consideration. In several studies, the existence of urban green areas is considered an amenity that could contribute to the increase in property values that are located in the vicinity of those areas [61].

The limitations of the current review can be mainly linked to the fact that, despite an effort to categorize evidence of the included studies, the findings in each category covered different aspects, expressed in different scales or monetary values. Therefore, it is hard to adopt a unified measurement approach or single values for the estimation of the same parameter on another site. In addition, a lack of self-reported data can be observed, and,

furthermore, in two specific categories, monetary and quality values, less evidence was found than in other categories. This was basically due to the methodology employed in those studies, which was more complex, in contrast with the other studies. Those studies utilized a more comprehensive analysis framework, thus investigating more than one important parameter. On the contrary, the other studies aimed at revealing more obvious and distinct values regarding green urban spaces, such as property values and carbon sequestration. However, this does not underestimate the added value that those studies might offer for further policy research.

The findings of the current study can be applied not only to existing urban spaces but also to the evaluation of future projects and urban policy design. Taking into consideration the pressure for various and differential land uses (commercial, residential, industrial), the evaluation of the benefits of urban green areas is necessary for a comprehensive consideration of the multiple values of urban green areas. In any case, an ex-ante study for the realization for an urban park is deemed necessary as it will provide measurable, and in several cases monetized, results and effects on the citizens and the local environment. By providing those values, they could serve as point of reference and can be compared with other land use development, which might have more measurable outcomes, e.g., the construction of an urban infrastructure that could result in the creation of a number of direct/indirect jobs and to revenues to the local authority.

A recent study [62] has stressed the importance of the green areas in urban spaces, focusing on their multi-functional role in a viable development of urban areas. The quality of life in big cities, as well as their overall visual concept, is relevant to the health and appearance of the trees found in parks and alleys. It has become more than obvious that the design of green infrastructure is one of the most important tools for the maintenance of health and recreation in cities. Urban green space managers and planners have to consider the benefits, the potential threats, and the management costs associated with urban greenery. Therefore, urban forestry plans should begin with an in-depth study and full consideration of the contribution made by urban green spaces and their importance for the quality of city life and human well-being.

## 5. Conclusions

The conducted review revealed a number of findings and recommendations that could direct future research and influence the design of urban policies. Those findings could assist local authorities as well as stakeholders to measure and assess the benefits of green spaces in urban parks and could promote interventions and programs to support their further deployment. By calibrating the methodology and adopting site-specific assumptions, this study's evidence can be further utilized to evaluate the impact of urban parks in future studies. Regarding future research, the relatively small number of included articles demonstrates the need for further research on overall socio-economic benefits in terms of co-values and enhanced urban liveability.

**Supplementary Materials:** The following are available online at https://www.mdpi.com/article/10 .3390/su13147863/s1, Table S1: Evidence Table of the Structured Review.

**Author Contributions:** Conceptualization, A.K., G.M., E.P. and A.D.S.; methodology, A.K., G.M. and E.P.; formal analysis, A.K., G.M. and E.P.; investigation, A.K., G.M. and E.P.; resources, A.K., G.M. and E.P.; data curation, A.K., G.M. and E.P.; writing—original draft preparation, A.K., G.M. and E.P.; writing—review and editing, A.K., A.D.S., N.P., E.V.A., E.K., G.K., K.T., G.M. and E.P.; visualization, A.K., K.T. and A.D.S.; supervision, A.K., A.D.S. and K.T.; funding acquisition, K.T. All authors have read and agreed to the published version of the manuscript.

**Funding:** The GrIn project "Promoting urban Integration of GReen INfrastructure to improve climate governance in cities" LIFE17 GIC/GR/000029 is co-funded by the European Commission under the Climate Change Action-Climate Change Governance and Information component of the LIFE Programme and the Greek Green Fund.

**Institutional Review Board Statement:** Not applicable.

**Informed Consent Statement:** Not applicable.

**Acknowledgments:** The authors highly acknowledge all the involved partners of the The GrIn project "Promoting urban Integration of GReen INfrastructure to improve climate governance in cities" LIFE17 GIC/GR/000029.

**Conflicts of Interest:** The authors declare no conflict of interest.

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
