# Peer review of "The Socioeconomic Welfare of Urban Green Areas and Parks; A Literature Review of Available Evidence"

_sustainability, doi:10.3390/su13147863_

Round 1
Reviewer 1 Report
Dear Authors,
Your article is interesting but has to be significantly improved.
I have the following remarks and suggestion for quality improvement of the article:
Why abstract in the article looks different than in the system of the journal? And why it is divided to separate parts? I think it is not needed to split the abstract to separate parts as methods, results, and conclusions.
I have a suggestion to extend the keywords section by adding the following terms: monetary values, CO₂ values. Instead, the keyword “urban parks” should be “urban green areas and parks”.
Due to the article-specific as “literature review of available evidence of the socio-economic welfare of Urban Green Areas and Parks”, the literature list should be expanded up to 50-60 literature sources as a minimum.
Page 4, row 114 provides the reference to (Table 1. Evidence Table), and Page 4, row 116-117 provides the reference to (Table 2. Summary Table). But in the article, there are no tables have been presented.
In section 3.3. Analysis of Evidence the authors presenting in each group of study types different countries. But it could be more logical to analyze more or less the same counties and their situation through all 5 groups.
I suggest analyzing the situation more or less in the same countries according to the five principal groups of study types: “willingness to pay (WTP)” property values; monetary values, CO₂ values, and qualitative criteria.
Conclusions should be based on research results and they don’t have to contain any references to the literature sources.
Wish you good luck
Author Response
- Why abstract in the article looks different than in the system of the journal? And why it is divided to separate parts? I think it is not needed to split the abstract to separate parts as methods, results, and conclusions.
Thank you for this comment. Changes have been accepted and captured as suggested in the revised abstract lines 9-31.
- I have a suggestion to extend the keywords section by adding the following terms: monetary values, CO₂ Instead, the keyword “urban parks” should be “urban green areas and parks”.
Thank you for this comment. Suggestions have been accepted and captured in the revised keywords, lines 32-33.
- Due to the article-specific as “literature reviewof available evidence of the socio-economic welfare of Urban Green Areas and Parks”, the literature list should be expanded up to 50-60 literature sources as a minimum.
Thank you very much for this comment. Please note that our initial goal was to present a structured review, following the PRISMA guidelines for systematic reviews and metanalysis. However, due to certain limitations, time constraints and difficulty for setting a systematic review protocol, we decided to present our study as literature review, since some important aspects of the systematic review were omitted. The number of 14 studies came out by employing the inclusion and exclusion criteria as described in the document. However, we agree that for literature review the number of 14 studies is quite limited. For this reason we agreed to expand our analysis by incorporating an extended review of 47 relevant papers coming up from our searches. This analysis is now presented in table 2 and also expanded as a text in the new section “3.3.1. Extended Literature Review”, lines 130-202. Please note, that we also considered to keep our initial structured analysis of the 14 included papers, since this has followed specific methodological steps which cannot be omitted and has led to important findings and policy recommendations already described in the manuscript.
- Page 4, row 114 provides the reference to (Table 1. Evidence Table), and Page 4, row 116-117 provides the reference to (Table 2. Summary Table). But in the article, there are no tables have been presented.
Thank you very much for this comment. Please note that we have now added two new tables in text. Specifically, and according to our revised analysis, the first table is Table 1. Summary Table of Structured Review, line 126 and the second table is Table 2. Summary Table of Extended Review, line 203. In addition, we have kept the detailed analysis of our 14 included studies as a supplementary Table S1. Evidence Table.
- In section 3. Analysis of Evidencethe authors presenting in each group of study types different countries. But it could be more logical to analyze more or less the same counties and their situation through all 5 groups.
Thank you very much for this comment. We would kindly like to bring to your attention that the initial split of analysis has been based on the categorization of the 14 included studies into the five main categories, also present into our S.1 Evidence Table. These categories are namely: willingness to pay (WTP)” property values; monetary values, CO₂ values, and qualitative criteria. Therefore, according to the different types of studies that these categories have been filled and accordingly the evidence is presented for the countries in which the studies have been implemented. We would rather emphasize the different methodological approaches and results that these approaches to our main research question. We really hope that this analysis is accepted by you.
- I suggest analyzing the situation more or less in the same countries according to the five principal groups of study types: “willingness to pay (WTP)” property values; monetary values, CO₂ values, and qualitative criteria.
Thank you very much for this comment. Similar to our comment above, we would like to point out again that the analysis of the 14 papers consisting our structured review has been based on the categorization of five main identified categories, also present into our S.1 Evidence Table.
- Conclusions should be based on research results and they don’t have to contain any references to the literature sources.
Thank you for this comment. Changes have been accepted and captured as suggested in the revised conclusion, lines 405-413.
Reviewer 2 Report
This manuscript is about a literature review to investigate the socioeconomic benefits derived by urban green spaces. However, the breadth and depth of literature covered in the manuscript are too narrow and shallow. This manuscript reviewed mainly 14 research arcticles under the rubric of the socioeconomic benefits of urban green space and tried to provide a short literature review. It is necessary that the literature reivew covers up more articles and is described more specifically. This manuscript missed a theme related to 'urban green space and health'. In addition, the manuscript should add up a theme concerning how the socicoeocnomic benefits of urban green spaces were measured.
What follows is other minor things to be revised.
- in line 18, the 'haspartly' needs to be changed to 'has partly'.
- in line 18, the 'PRISMA' should be spelled out.
- in line 59 and line 61, the comma(,) should be eliminated.
Author Response
- This manuscript is about a literature review to investigate the socioeconomic benefits derived by urban green spaces. However, the breadth and depth of literature covered in the manuscript are too narrow and shallow. This manuscript reviewed mainly 14 research arcticles under the rubric of the socioeconomic benefits of urban green space and tried to provide a short literature review. It is necessary that the literature reivew covers up more articles and is described more specifically.
Thank you very much for this comment. Please note that our initial goal was to present a structured review, following the PRISMA guidelines for systematic reviews and metanalysis. However, due to certain limitations, time constraints and difficulty for setting a systematic review protocol, we decided to present our study as literature review, since some important aspects of the systematic review were omitted. The number of 14 studies came out by employing the inclusion and exclusion criteria as described in the document. According also to your comment, we agreed to expand our analysis by incorporating an extended review of 47 relevant papers coming up from our searches. This analysis is now presented in table 2 and also expanded as a text in the new section “3.3.1. Extended Literature Review”, lines 130-202.
- This manuscript missed a theme related to 'urban green space and health'.
Thank you very much for this comment. Please note that a special reference on “urban green space and health” has been now provided in lines 138-165.
- In addition, the manuscript should add up a theme concerning how the socioeconomic benefits of urban green spaces were measured.
Thank you very much for this comment. Please note note that we have now added in text Table 1. Summary Table of Structured Review, line 126, which provides an analysis of how socioeconomic benefits were categorized and measured according to the 14 included articles of the structured review. In addition, an overall analysis of different benefit levels is also provided in new section “3.3.1. Extended Literature Review”, lines 130-202.
- What follows is other minor things to be revised.
- in line 18, the 'haspartly' needs to be changed to 'has partly'.
- in line 18, the 'PRISMA' should be spelled out.
- in line 59 and line 61, the comma(,) should be eliminated.
Thank you very much for this comment. All the suggested changes have been now addressed.
Reviewer 3 Report
The topic covered by this paper is an interesting and current one. However, there are some concerns, mostly related to the method and approach used:
1) There is no explanation why GoogleScholar and Scopus were selected
2) There is no proper description of the queries used in those databases. Authors provide the list of terms: “Urban green space” OR “Urban green areas” OR 84 “Urban parks” OR “Urban ecosystems” AND “economic value” OR “economic benefits” OR 85 “ecosystem services” OR “monetary values” OR “carbon monetary values” OR “climate change” - however, was it limited to the title field? Was it a full-text search? Was it a subject search? And this is important information in the case of the Scopus database.
3) the First stage of the research process included 1560 records without replicates, yet somehow the review is based on 14 papers. In situations like this, it is safe to assume that the selection criteria (queries) used were not properly defined and/or the following selection steps were too strict. Those 14 papers are less than 0,9% of the records collected from the Scopus and GS.
4) Finally, 14 papers are not enough to make proper and valid conclusions. As the result, the reviewed paper is a summary of those records.
Author Response
1) There is no explanation why Google Scholar and Scopus were selected
Thank you very much for this comment. Google Scholar and Scopus were used as being two major databases with freely-accessible articles related to socioeconomic analysis, as also added in lines 86-88.
2) There is no proper description of the queries used in those databases. Authors provide the list of terms: “Urban green space” OR “Urban green areas” OR 84 “Urban parks” OR “Urban ecosystems” AND “economic value” OR “economic benefits” OR 85 “ecosystem services” OR “monetary values” OR “carbon monetary values” OR “climate change” - however, was it limited to the title field? Was it a full-text search? Was it a subject search? And this is important information in the case of the Scopus database.
Thank you very much for your comments. Please find some further explanation as required in lines 86-88, 103-106, 117-122.
3) The First stage of the research process included 1560 records without replicates, yet somehow the review is based on 14 papers. In situations like this, it is safe to assume that the selection criteria (queries) used were not properly defined and/or the following selection steps were too strict. Those 14 papers are less than 0,9% of the records collected from the Scopus and GS.
Thank you very much for this comment. Please note that our initial goal was to present a structured review, following the PRISMA guidelines for systematic reviews and metanalysis. However, due to certain limitations, time constraints and difficulty for setting a systematic review protocol, we decided to present our study as literature review, since some important aspects of the systematic review were omitted. The number of 14 studies came out indeed by employing the inclusion and exclusion criteria as described in the document. However, we agree that for literature review the number of 14 studies is quite limited. For this reason we considered to expand our analysis by incorporating an extended review of a total 47 relevant papers coming up from our searches. This analysis is now presented in table 2 and also expanded as a text in the new section “3.3.1. Extended Literature Review”, lines 146-220. Please note, that we also considered to keep our initial structured analysis of the 14 included papers, since this has followed specific methodological steps which cannot be omitted and has led to important findings and policy recommendations already described in the manuscript.
4) Finally, 14 papers are not enough to make proper and valid conclusions. As the result, the reviewed paper is a summary of those records.
Thank you very much for this comment. Please note that in line with the reply to the comment above, we have now added two new tables in text. Specifically, and according to our revised analysis, the first table is Table 1. Summary Table of Structured Review, line 141 and the second table is Table 2. Summary Table of Extended Review, line 220. In addition, we have kept the detailed analysis of our 14 included studies as a supplementary Table S1. Evidence Table.
Round 2
Reviewer 1 Report
The authors improved the article according to the comments.
The article can be accepted for publication.
Author Response
Dear Reviewer, thank you so much for your comments, very highly appreciated.
Reviewer 2 Report
The revised manuscript was much improved according to reviewers' comments.
However, a couple of minor things should be revised before acceptance.
- In Tables 1 and 2, the term 'observational-analytic(OA)' under the study type field is unclear. Please explain fully what this term indicates.
- In Lines 206, 220, 249, 265 and 272, the title number should be changed like 3.3.1.x.
- In Abstract, the summary of an extended literature review should be reflected.
Author Response
- In Tables 1 and 2, the term 'observational-analytic(OA)' under the study type field is unclear. Please explain fully what this term indicates.
Thank you very much for this comment. An explanation of the term Observational Analytic, has been now provided in lines 138-142.
- In Lines 206, 220, 249, 265 and 272, the title number should be changed like 3.3.1.x.
Thank you for this comment. The title numbering has been revised according to your comment in lines 223, 237, 267, 283, 290.
- In Abstract, the summary of an extended literature review should be reflected.
Thank you very much for this comment. A reflection of the extended literature review has been now added in lines 24-27 of the abstract.